# Diverse patients' attitudes towards Artificial Intelligence (AI) in diagnosis

**Christopher Robertson**[1,2], **Andrew Woods**[1], **Kelly Bergstrand**[3], **Jess Findley**[1], **Cayley Balser**[1], **Marvin J. Slepian**[1] *

1 University of Arizona, Tucson, Arizona, United States of America, 2 Boston University, Boston, Massachusetts, United States of America, 3 University of Texas at Arlington, Arlington Texas, United States of America

* slepian@arizona.edu

**Data Availability Statement:** Analysis was done in Stata 17. Data and code are publicly available on the Open Science Framework, osf.io/9y26x.

**Funding:** This study was funded by the National Institutes of Health (3R25HL126140-05S1 to CR

## Abstract

Artificial intelligence (AI) has the potential to improve diagnostic accuracy. Yet people are often reluctant to trust automated systems, and some patient populations may be particularly distrusting. We sought to determine how diverse patient populations feel about the use of AI diagnostic tools, and whether framing and informing the choice affects uptake. To construct and pretest our materials, we conducted structured interviews with a diverse set of actual patients. We then conducted a pre-registered (osf.io/9y26x), randomized, blinded survey experiment in factorial design. A survey firm provided n = 2675 responses, oversampling minoritized populations. Clinical vignettes were randomly manipulated in eight variables with two levels each: disease severity (leukemia versus sleep apnea), whether AI is proven more accurate than human specialists, whether the AI clinic is personalized to the patient through listening and/or tailoring, whether the AI clinic avoids racial and/or financial biases, whether the Primary Care Physician (PCP) promises to explain and incorporate the advice, and whether the PCP nudges the patient towards AI as the established, recommended, and easy choice. Our main outcome measure was selection of AI clinic or human physician specialist clinic (binary, "AI uptake"). We found that with weighting representative to the U.S. population, respondents were almost evenly split (52.9% chose human doctor and 47.1% chose AI clinic). In unweighted experimental contrasts of respondents who met pre-registered criteria for engagement, a PCP's explanation that AI has proven superior accuracy increased uptake (OR = 1.48, CI 1.24–1.77, $p < .001$), as did a PCP's nudge towards AI as the established choice (OR = 1.25, CI: 1.05–1.50, $p = .013$), as did reassurance that the AI clinic had trained counselors to listen to the patient's unique perspectives (OR = 1.27, CI: 1.07–1.52, $p = .008$). Disease severity (leukemia versus sleep apnea) and other manipulations did not affect AI uptake significantly. Compared to White respondents, Black respondents selected AI less often (OR = .73, CI: .55-.96, $p = .023$) and Native Americans selected it more often (OR: 1.37, CI: 1.01–1.87, $p = .041$). Older respondents were less likely to choose AI (OR: .99, CI: .987-.999, $p = .03$), as were those who identified as politically conservative (OR: .65, CI: .52-.81, $p < .001$) or viewed religion as important (OR: .64, CI: .52-.77, $p < .001$). For each unit increase in education, the odds are 1.10 greater for selecting an AI provider (OR: 1.10, CI: 1.03–1.18, $p = .004$). While many patients appear resistant to the use of AI, accuracy information, nudges and a listening patient experience

and AW). The funders had no role in study design, data collection and analysis, decision to publish, or preparation of the manuscript.

**Competing interests:** The authors have declared that no competing interests exist.

may help increase acceptance. To ensure that the benefits of AI are secured in clinical practice, future research on best methods of physician incorporation and patient decision making is required.

## Introduction

Artificial intelligence (AI) is poised to transform healthcare. Today, AI is used to analyze tumors in chest images [1], regulate implanted devices [2], and select personalized courses of care [3]. Despite the promise of AI, there is broad public skepticism about AI in a range of domains from transportation to criminal justice to healthcare [4,5]. Doctors and patients tend to primarily rely on doctors' clinical judgment, even when it is at odds with statistical judgment [6].

Research shows that patients prefer human doctors to AI-powered machines in diagnosis, screening, and treatment [7–10]. In an early study, patients were more likely to follow medical advice from a physician than a computer and were less trustful of computers as providers of medical advice [7]. Other work shows that patients are less trusting of doctors that rely on non-human decision aids [8,9]. More recently, in a series of studies, patients were less willing to schedule an appointment to be diagnosed by a robot, and they were willing to pay significantly more money for a human provider, with a reported perception that AI providers are less able to account for patients' unique characteristics [10].

Yet acceptance of AI may depend on specific features of the system and how the choice is framed, and there may be differences among groups of patients [11]. Outside of healthcare, consumers have been shown to more often trust AI systems for objective tasks, while subjective tasks are viewed as more appropriate for humans [12]. Some qualitative research suggests that lower levels of patient education are associated with lower trust in computerization [13]. Other small studies suggest that AI may be acceptable if human physicians ultimately remain in control [14]. And, although patients may prefer human physicians, some work suggests that they may better adhere to advice coming from algorithms [15].

More generally, research suggests that patients' trust in their physicians is an essential component of effective healing [16,17]. Black, Hispanic, and Native Americans reportedly have lower levels of trust in their physicians [18,19,20]. These communities have experienced harm historically from components of the medical system (e.g., the Tuskegee Syphilis Study). Yet trust can be enhanced when patients and their providers have similar background, geography, or ethnic groups [21,22,23]. Patients may be concerned about human physicians being biased by their financial relationships with pharmaceutical companies [24] or biased by implicit racial stereotypes [25]. Although initial forays into algorithmic decisions gave rise to similar concerns [26], AI systems may be rigorously designed, tested, and continuously monitored to minimize racial or financial biases in healthcare [27,28].

There are many drivers to the development and uptake of AI in healthcare, including commercial incentives and physician attitudes [29,30]. Trustworthiness may depend on the relationship established between users, infrastructures, technologies, and practitioners, rather than the certainty and accuracy of the technology [31,32].

To the extent that patients retain the right of informed consent with regard to their own healthcare, much will depend on patient attitudes towards AI. To that end, we used qualitative and quantitative methods to study diverse patient populations' views about AI in medicine.

To our knowledge, this is the first large-scale population-based survey experiment, with random assignment to realistic clinical vignettes systematically manipulated to analyze a range

of factors that could influence AI uptake with patients. Moreover, our study is enriched to allow sufficient sample size to compare AI uptake across five different racial/ethnic groups, including those who have historically shown lower levels of trust in the healthcare system.

## Methods

We conducted two study phases, one qualitative and one quantitative. In the qualitative phase (February to December 2020), we conducted structured interviews with 24 patients recruited for racial and ethnic diversity to understand their reactions to current and future AI technologies. In the quantitative phase (January and February 2021), we used an internet-based survey experiment oversampling Black, Hispanic, Asian, and Native American populations. Both phases placed respondents as mock patients into clinical vignettes to explore whether they would prefer to have an AI system versus a doctor for diagnosis and treatment and under what circumstances.

We chose this mixed-methods design for a few reasons. First, because this is the first study of its kind, we wanted to ensure that the vignettes driving our quantitative survey were realistic and intuitive; the qualitative pre-study helped us to gauge participant reaction. Second, large-scale quantitative surveys often raise a number of questions about why people respond the way they do. The mixed-method design allows us to accomplish something that neither approach —purely quantitative nor purely qualitative—would achieve on its own.

### Development and oversight of the survey

To develop clinical vignettes, we consulted physicians specializing in cardiology, pulmonology, hematology, and sleep medicine to develop vignettes, which were reviewed by physician co-author (MJS), for authenticity. This study was determined to be exempt by the Human Subjects Protection Program (Institutional Review Board) at the University of Arizona, and all subjects consented.

### Qualitative pre-study

Our qualitative study pre-tested the vignettes and generated hypotheses. For 30–60 minute qualitative interviews in Spanish or English, we recruited 24 individuals from clinics in Tucson, Arizona, including 10 White, 8 Hispanic, 3 Black, 2 Native American, and 1 Asian patients. In total, 16 were females and 8 were males. Ages ranged from 19 to 92 years, with most over 50, as would be expected given our recruitment from a cardiac clinic. Educational achievement was relatively high, with 7 of the subjects having a graduate degree, and 5 having a bachelor's degree.

The nature of these interviews was semi-structured. All participants were given the same script describing the topic, the research design, and the sample vignettes. The prompts were deliberately open-ended to allow participants to share reactions and feedback to make sure the vignettes were understood. We recorded all of our interviews and had them transcribed to better inform the design of the quantitative survey.

After getting informed consent, each interview began with an open-ended question asking the participant to think back to a difficult medical decision, and in particular "who or what influenced your decision?" This prompted a wide array of responses. The most common influence on participants' decision-making was their primary care physician, though several also noted family and friends influencing their decision. We asked another open-ended question: "generally how do you feel about doctors relying on computer systems to make treatment decisions for you?" This prompted a broad array of responses, with some participants expressing fear or anxiety (and occasionally humor) about the increasing use of machines in everyday life. We often probed to

distinguish routine use of electronic health records (EHRs) systems, which were quite familiar to respondents, versus computerized diagnostic tools, which were less familiar.

The core of our interviews were the vignettes which asked participants to imagine themselves in particular medical scenarios. We started by asking participants to imagine their primary care physician recommending a change to their diet and exercise, based on a family history of leukemia and advice from "the MedX computer system with data from millions of other patients." This provoked a mild reaction, with most participants noting they were already being told to mind their diet and exercise habits. Then we raised the stakes. Participants were told to suppose they start to feel tired and achy, and their primary care physician wants a second opinion from either an oncologist or a new AI-driven lab that "is more accurate than the oncologist." The participants were asked whether they'd choose one or the other, or pay $1000 to visit both. Tellingly, the majority of patients said they'd prefer to see only the oncologist, despite being told the oncologist was less accurate than the AI lab.

We presented another vignette involving sleep apnea, where participants were asked whether they'd rather visit a traditional sleep clinic requiring an overnight stay away from home and interpretation by a human physician versus an at-home device that relies on self-placed sensors and AI diagnostic interpretation. We saw a broad range of views in response to this vignette, with several participants having strong and perhaps idiosyncratic reactions based on their personal experiences dealing with sleep apnea and visiting sleep clinics.

Overall, while some patients expressed confidence that AI systems could achieve greater accuracy in diagnosis and treatment compared to a physician, several patients called on their own experiences with technology to suggest that an AI system could be fallible. Other patients, especially those who were non-White, expressed lack of trust with the healthcare system more generally and recounted anecdotes where they felt unheard or mistreated. Patients nearly uniformly said they would rely heavily on their physicians to guide their choice of whether an AI system would be used for their diagnosis or treatment, but most nonetheless emphasized that they would generally want to know of such use, suggesting that it is material for their informed consent. Several patients indicated that they had greater confidence in their human physicians than an AI system to personalize treatment decisions to the patient's own situation. The technology was more attractive for younger and more educated patients. Several patients invoked their belief in God as being important to their healthcare decisions, and some suggested confidence that God would work through human physicians.

## Quantitative survey experimental design and materials

We designed our quantitative phase as a blinded, randomized survey experiment in factorial design, manipulating eight variables by substituting text in or out of the base vignette. Respondents were block randomized by race/ethnicity to experimental conditions. We counterbalanced whether respondents answered certain covariate questions before or after our vignettes and primary outcomes.

The full text of the vignettes and manipulations are shown at osf.io/9y26x. These materials were based on the vignettes that we tested in the qualitative phase, with refinements and clarifications based on feedback from those participants. The base clinical vignette was split into two parts, with an initial segment laying out the patient's history and primary care physician's (PCP's) initial impressions. As one of the experimental manipulations, all respondents saw either a leukemia or a sleep apnea version of the base case, with the PCP explaining that leukemia could be fatal if not properly diagnosed and treated, while sleep apnea was described as interfering with the patient's comfort and lifestyle. Respondents were asked to explain their "reactions and feelings at this point in the story."

**Table 1. Presentation of Provider Choice.**

| Text Presented to Respondents |
| --- |
| "Your doctor offers either of two options:<br>• You could go to the offices of Dr. Williams, [a hematologist-oncologist / NA], a specialist doctor who is trained to diagnose [leukemias / sleep disorders], or<br>• You could go to the office of Med-X, which is built around a proprietary computer system designed to diagnose [leukemias / sleep disorders]. Your blood and genetic information would be drawn by a nurse, and the medical analysis would be done entirely by a machine using artificial intelligence (AI). With every case it sees from tens of thousands of patients worldwide, the AI system gets more accurate in its diagnoses. If you visit the AI clinic, your data will be de-identified and then become part of the system.<br>[accuracy manipulation] Your insurance will only cover one of the services so you must choose which one to use to diagnose your possible [sleep apnea / leukemia]. Your out-of-pocket cost is the same either way. [other manipulations]" |

A final segment of the vignette explained that, "your doctor would like to get a second opinion on whether you have leukemia, and if so get the best treatment plan," or "to determine whether you actually have an apnea, determine its type, and determine the best course of treatment, your physician suggests a sleep study." The PCP then presented the choice of AI versus physician specialist, followed by the experimental manipulations. Table 1 displays the description of the two providers. Table 2 shows a summary of the manipulations in either Level 1 or Level 2, which followed this presentation. In several of the manipulations, when Level 1 was randomly selected, the vignettes were simply silent about the issue.

**Table 2. Experimental Conditions.**

| Variable | Level 1 | Level 2 |
| --- | --- | --- |
| Illness Severity | Leukemia: ". . .There are several different types of leukemia. Some . . . can spread to lymph nodes or the central nervous system if not treated. Leukemia can be fatal. . .") | Sleep apnea: ". . .you are getting poor sleep at night, with very loud snoring, and sometimes it seems like you stop breathing or gasp for air during sleep. . . . |
| AI Accuracy | Silent: No description of this issue provided in the vignette. | "Your doctor tells you that, based on scientific studies in leading journals, the AI system is proven more accurate at diagnoses compared to even specialist human physicians." |
| AI Listens (Personal) | Silent: No description of this issue provided in the vignette. | "The Med-X clinic staff will carefully listen to understand your lifestyle, preferences, values, and goals. . . . you will have an extensive 45-minutes interview with a trained counselor, who will ask a range of questions to get your perspective on your healthcare." |
| AI Tailored (Personal) | Silent: No description of this issue provided in the vignette. | "The AI system's advice will be tailored to you. [It] will incorporate 36 different measurements and attributes specific to you to generate a unique and personalized treatment plan, just for you." |
| AI Racial Unbiased | Silent: No description of this issue provided in the vignette. | "Although research suggests that human physicians can be biased by racial and ethnic stereotypes, the AI system has been carefully designed and tested to ensure that treatment recommendations are unbiased." |
| AI Financial Unbiased | Silent: No description of this issue provided in the vignette. | "Although research suggests that human physicians can be biased by their financial relationships with drugmakers and insurance companies, the AI system has been carefully designed and tested to ensure that treatment recommendations are unbiased." |
| PCP Incorporates AI Advice | Defer-Portal: "Your doctor explains, 'From either clinic, you will receive the results as an electronic message in your patient portal, the next day. It will tell you the diagnosis and what to do next.'" | Incorporate-Explain: "Your doctor explains, "When we get the results back from either clinic, I will explain them and talk them through with you. I will incorporate the results into my ultimate opinion on what we should do next.'" |
| PCP Nudges toward AI | None: "Your doctor says, 'We can get you into either Dr. Williams or the Med-X AI clinic; it is your choice.'" | Default-Easy: "Your doctor says, 'For some time, I have been recommending the Med-X AI clinic for all my patients, and the nurse has already confirmed available appointments for you. But if you prefer to see Dr. Williams, I can give you a referral instead. It is your choice.'" |

**Note:** For the unbiased and personalization variables, when both types (race and financial, or listens and tailored) were presented, the text was concisely integrated into a single statement, as shown in the survey text at osf.io/9y26x.

Our primary outcome variable ("AI uptake") was binary, "Which provider would you choose to diagnose your health problem? Dr. Williams, the specialist physician [or] The Med-X clinic, the AI computer system," with presentation order randomized. Finally, we presented several debriefing questions. These included an attention-check question to test whether the respondent could identify the disease featured in the vignette read a few minutes prior and a self-assessment of the respondent's understanding of the vignette, on a ten-point scale.

## Survey administration and screening

YouGov interviewed 2875 U.S. adults who identified as White, Black, Latino, Asian or Native American, who were then matched down to a sample of 2675 to produce the final dataset. To allow well-powered estimates for particular racial and ethnic groups, the sample was designed to over-represent non-White Americans as follows: 650 Whites (24%), 575 Blacks (21%), 550 Latinos (21%), 550 Asians (21%), and 350 Native Americans (13%). YouGov's matching and weighting approach is described in S1 Text.

We report both weighted and unweighted analyses. This allows us to both represent the United States population as well as investigate differences by racial groups and those who fully engaged with the vignettes. For the unweighted analyses we exclude respondents based on some of the previously established criteria in the pre-registration plan: those that failed an attention check about whether respondent could correctly recall the disease presented in the vignette, and those reporting that they did not understand the vignette (bottom two levels on 10-point scale). After these removals, the sample for our primary analyses was N = 2,472. We present models without exclusions in S1 Table and S2 Table.

## Characteristics of the respondents

Our sample was diverse and representative after weights are applied. As shown in Tables 3 and 4, in addition to race/ethnicity coverage, respondents span multiple educational levels, with roughly equal amounts of respondents in the categories of high school degree or less, some college or a two-year degree, and a four-year college degree or higher. We have slightly more females in the sample. The average age was 48 years, and about half made less than $50,000 a year (N = 1194; 54%). Nearly one-quarter of respondents (23.27%) identified as political conservatives, as the top-two levels on a standard six-point scale ("very liberal", "liberal", "moderate", "conservative", or "very conservative", with respondents also allowed to say "not sure" or skip the question). Nearly two-thirds of respondents (62.14%) viewed religion as important, as measured by the top two levels on a four-point Likert scale ("very important", "somewhat important", "not too important", or "not at all important"). Table 2B disaggregates these descriptive statistics by race and ethnic groups.

## Statistical analysis

Our primary analyses rely on multivariable logistic regressions in STATA, and the computation of predictive margins and 95% confidence intervals (CI), exploiting the randomized design. We use $\alpha = .05$ as the threshold for significance of p-values.

To generate estimates representative of the U.S. population, we conducted analyses with weighted data, without exclusions (N = 2672). Here we use the Stata svy suite of commands to estimate the proportions shown in Fig 1 and the text. Combined, these analyses can both speak to our specific interest in diverse populations as well as reflect larger trends in the U.S. population.

In Table 5, Model 1 merely shows effects of our experimental conditions controlling for whether covariates were collected before or after the vignettes (order) and familiarity or

**Table 3. Descriptive Traits of Participants (N = 2472).**

| Factor | Mean (SD) or N (%) |
| --- | --- |
| Age | 48.07 (17.18) |
| Female | 1380 (55.83%) |
| Married | 1183 (47.86%) |
| Employed Full-Time | 853 (34.51%) |
| Less than 50K Income | 1194 (54.27%) |
| High School or Less | 864 (34.95%) |
| Some College or Associate Degree | 777 (31.43%) |
| Bachelor's Degree or More | 831 (33.61%) |
| Conservative* | 575 (23.27%) |
| Religion Important* | 1536 (62.14%) |
| White | 617 (24.96%) |
| Black | 528 (21.36%) |
| Hispanic | 489 (19.78%) |
| Asian | 511 (20.67%) |
| Native American | 327 (13.23%) |
| Good Health* | 1897 (76.74%) |
| Have Provider* | 2001 (80.95%) |
| Trust Provider* | 1956 (79.13%) |
| Trust Hospital* | 1735 (70.21%) |
| Trust AI Companies* | 1346 (54.45%) |

**Note:** Factors shown with asterisk (*) are based on groupings in Likert-scales. For example, good health (1 = excellent/very good/good), religion viewed as important (1 = very/somewhat important), and conservative (very conservative/conservative). The trust measures (provider/hospital/AI companies) were grouped by selections of 1–5 on a -5 (complete distrust) to 5 (complete trust) scale. Income was measured on 16-point scale (from less than $10,000 to more than $500,000), education was measured on a 6-point scale (from less than high school to graduate school).

experience with the illness. Model 2 adds demographic controls shown in the Table, although the sample size is smaller primarily due to missing income data. Model 3 also controls for respondents' attitudes regarding trust in providers, hospitals and AI companies, and these estimates and p-values are displayed in Fig 2 and discussed in the text, except where otherwise noted. Wald tests of model fit find significant improvement moving from Model 1 to Model 2 [$\chi^2$ (16, N = 2199) = 87.15; $p < .001$] as well as moving from Model 2 to Model 3 [$\chi^2$(3, N = 2198) = 106.75; $p < .001$]. We then analyze the final model in racial subsets in Table 6. The logistic regression models use unweighted data, exclude respondents who failed the manipulation checks, and use listwise deletion for missing data.

## Results

We found a substantial resistance to artificial intelligence. With weighting representative to the U.S. population, most respondents (52.9%) chose the human doctor and 47.1% chose AI clinic, with some variation along race and ethnicity, as shown in Fig 1.

### Effects of experimental manipulations

As shown in Table 5 Model 3, when AI was proven to be more accurate, respondents were substantially more likely to select it; this was one of the most pronounced effects (OR = 1.48, CI

**Table 4. Descriptive Traits of Participants (Mean (SD) or N (%)), Split by Race / Ethnicity (N = 2472).**

| Factor | White | Black | Hispanic | Asian | Native American |
|---|---|---|---|---|---|
| Age | 51.43 (17.86) | 49.42 (16.63) | 44.69 (16.36) | 43.63 (16.70) | 51.54 (16.35) |
| Female | 314 (50.89%) | 302 (57.20%) | 290 (59.30%) | 281 (54.99%) | 193 (59.02%) |
| Married | 315(51.05%) | 191 (36.17%) | 253 (51.74%) | 248 (48.53%) | 176 (53.82%) |
| Employed Full-Time | 218 (35.33%) | 165 (31.25%) | 172 (35.17%) | 210 (41.10%) | 88(26.91%) |
| Less than 50K Income | 271 (50.00%) | 332 (69.17%) | 253 (57.24%) | 157 (35.93%) | 181 (60.54%) |
| High School or Less | 207 (33.55%) | 228(43.18%) | 247(50.51%) | 90(17.61%) | 92(28.13%) |
| Some College or Associate Degree | 189 (30.63%) | 178 (33.71%) | 143 (29.24%) | 118 (23.09%) | 149 (45.57%) |
| Bachelor's Degree or More | 221 (35.82%) | 122 (23.10%) | 99 (20.25%) | 303 (59.30%) | 86 (26.30%) |
| Conservative* | 183 (29.66%) | 67 (12.69%) | 110 (22.49%) | 91 (17.81%) | 124 (38.04%) |
| Religion Important* | 331(53.65%) | 399(75.57%) | 337 (68.92%) | 253 (49.51%) | 216 (66.06%) |
| Good Health* | 468 (75.85%) | 398 (75.38%) | 379 (77.51%) | 415 (81.21%) | 237 (72.48%) |
| Have Provider | 509 (82.50%) | 416 (78.79%) | 372 (76.07%) | 414 (81.02%) | 290 (88.69%) |
| Trust Provider* | 492 (79.74%) | 409 (77.46%) | 372 (76.07%) | 414 (81.02%) | 269 (82.26%) |
| Trust Hospital* | 438 (70.99%) | 351 (66.60%) | 346 (70.76%) | 373 (72.99%) | 227 (69.42%) |
| Trust AI Companies* | 305 (49.42%) | 301 (57.01%) | 282 (57.67%) | 312 (61.06%) | 143 (43.73%) |
| N | 617 | 528 | 489 | 511 | 327 |

**Note:** Factors shown with asterisk (*) are based on groupings in Likert-scales. For example, good health (1 = excellent/very good/good), religion viewed as important (1 = very/somewhat important), and conservative (very conservative/conservative). The trust measures (provider/hospital/AI companies) were grouped by selections of 1–5 on a -5 (complete distrust) to 5 (complete trust) scale. Income was measured on 16-point scale (from less than $10,000 to more than $500,000), education was measured on a 6-point scale (from less than high school to graduate school).

1.24–1.77, $p < .001$). This effect arises from the manipulation that "Your doctor tells you that, based on scientific studies in leading journals, the AI system is proven more accurate at diagnoses compared to even specialist human physicians."

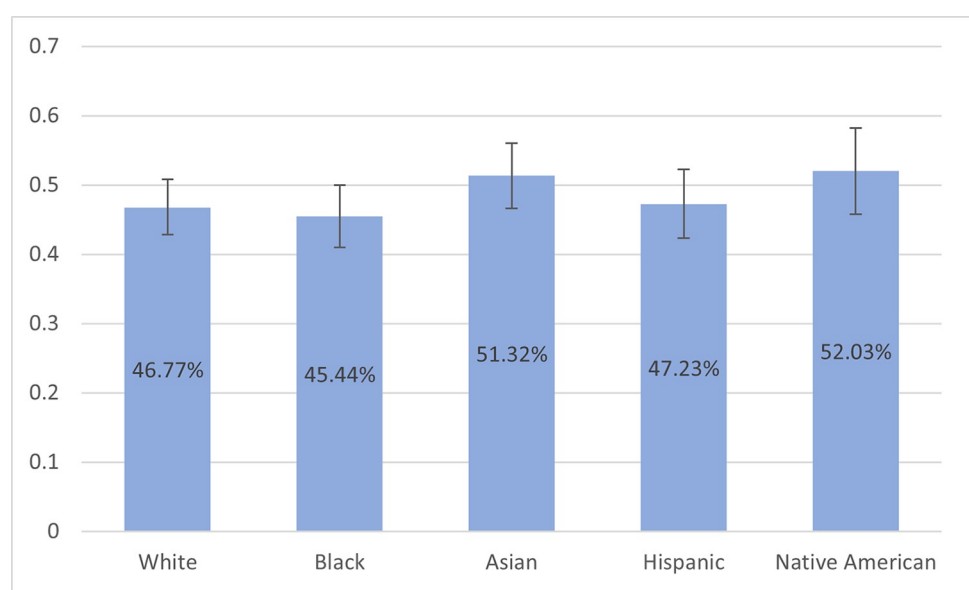

**Fig 1. Proportion of Respondents Selecting an AI Provider with 95% Confidence Intervals by Respondent Race / Ethnicity (N = 2672, Weighted to Represent U.S. Population). Notes:** Using the 2018 American Community Survey as the reference, the sample was weighted age, gender, race/ethnicity, years of education, and region to produce estimates for the U.S. population.

**Table 5. Logistic Regression Predicting AI Provider Choice Versus Human Physician.**

| | Model 1: Experimental Conditions | | Model 2: + Demographics | | Model 3: + Trust | |
|---|---|---|---|---|---|---|
| Illness Severity | 0.96 | (0.08) | 1.00 | (0.09) | 0.98 | (0.09) |
| Accuracy | 1.51*** | (0.12) | 1.54*** | (0.14) | 1.48*** | (0.13) |
| Incorporation | 0.92 | (0.07) | 0.91 | (0.08) | 0.88 | (0.08) |
| Established | 1.28** | (0.10) | 1.28** | (0.11) | 1.25* | (0.11) |
| Tailored | 1.06 | (0.09) | 1.07 | (0.09) | 1.11 | (0.10) |
| Listens | 1.20* | (0.10) | 1.23* | (0.11) | 1.27** | (0.12) |
| Race Unbiased | 1.08 | (0.09) | 1.07 | (0.09) | 1.10 | (0.10) |
| Financial Unbiased | 1.13 | (0.09) | 1.12 | (0.10) | 1.10 | (0.10) |
| Order | 1.20* | (0.10) | 1.20* | (0.11) | 1.13 | (0.10) |
| Illness Experience | 1.12 | (0.10) | 1.18+ | (0.11) | 1.17 | (0.12) |
| Black | | | 0.84 | (0.11) | 0.73* | (0.10) |
| Hispanic | | | 1.00 | (0.14) | 0.88 | (0.12) |
| Asian | | | 0.94 | (0.13) | 0.85 | (0.12) |
| Native | | | 1.26 | (0.19) | 1.37* | (0.21) |
| Income | | | 1.02 | (0.02) | 1.02 | (0.02) |
| Age | | | 0.99* | (0.00) | 0.99* | (0.00) |
| Conservative | | | 0.61*** | (0.07) | 0.65*** | (0.08) |
| Religion Important | | | 0.67*** | (0.06) | 0.64*** | (0.06) |
| Sex | | | 1.00 | (0.09) | 0.98 | (0.09) |
| Education | | | 1.07* | (0.04) | 1.10** | (0.04) |
| Married | | | 0.96 | (0.09) | 0.97 | (0.10) |
| Employed Full-Time | | | 1.03 | (0.10) | 1.05 | (0.11) |
| Good Health | | | 1.09 | (0.12) | 0.99 | (0.11) |
| Have Provider | | | 1.12 | (0.13) | 1.08 | (0.14) |
| Trust Provider | | | | | 0.96 | (0.03) |
| Trust Hospital | | | | | 0.95+ | (0.03) |
| Trust AI Companies | | | | | 1.29*** | (0.03) |
| N | 2471 | | 2199 | | 2198 | |

Notes: Odds Ratios; Standard errors in parentheses

+ $p < .10$

* $p < .05$

** $p < .01$

*** $p < .001$. For race, Whites are the reference group. Income (16-point scale from less than $10,000 to more than $500,000), age (years), education (6-point scale from less than high school to graduate school), sex (1 = male, 0 = female). Likert-scales are coded as good health (1 = excellent/very good/good), religion viewed as important (1 = very/somewhat important), and conservative (very conservative/conservative). Analysis excludes respondents that failed an attention check about whether respondent could correctly recall the disease presented in the vignette, and those reporting that they did not understand the vignette (bottom two levels on 10-point scale).

When the PCP nudged the patient toward AI as the established option, patients were more likely to choose it (OR = 1.25, CI: 1.05–1.50, $p$ = .013). This effect arises from the PCP saying that "For some time, I have been recommending the Med-X AI clinic for all my patients, and the nurse has already confirmed available appointments for you."

The results also show greater AI uptake when the AI system is personalized to listen to the patient (OR = 1.27, CI: 1.07–1.52, $p$ = .008). This effect is caused by the text manipulation: "The Med-X clinic staff will carefully listen to understand your lifestyle, preferences, values,

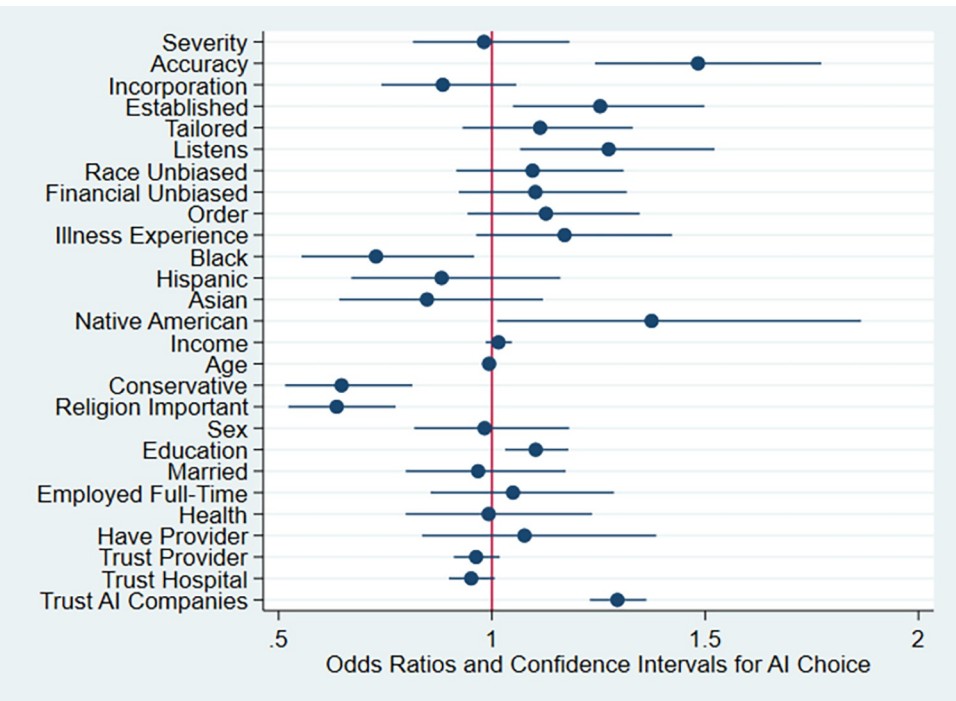

**Fig 2. Odds of Selecting an AI Provider by Experimental Condition, Demographics, and Attitudinal Variables, With 95% Confidence Intervals (N = 2198).** *Notes*: Analysis excludes respondents that failed an attention check requiring recall of the disease presented in the vignette, and those reporting that they did not understand the vignette (bottom two levels on 10-point scale). See Table 3, Model 3 for details.

and goals. At the Med-X clinic, you will have an extensive 45-minutes interview with a trained counselor, who will ask a range of questions to get your perspective on your healthcare."

Other experimental conditions did not have significant effects, such as illness severity, whether the doctor would merely defer to the results or incorporate them into his or her opinion, whether the AI system would be racially or financially unbiased, and whether the AI used multiple measurements to tailor the treatment plan (all $p > .05$).

## Demographic and attitudinal associations

Also shown in Table 5 Model 3, several demographic and attitudinal variables were also associated with AI uptake. Older respondents had significantly lower odds of choosing AI (OR: .99, CI: .987-.999, $p = .03$), which is modelled as a year-by-year effect. Conservatives have 35% less than equal odds than non-conservatives of choosing AI (OR: .65, CI: .52-.81, $p < .001$) and for each unit increase in education, the odds are 1.10 greater for selecting an AI provider (OR: 1.10, CI: 1.03–1.18, $p = .004$). Respondents who viewed religion as important had significantly lower odds of choosing AI (OR: .64, CI: .52-.77, $p < .001$). In terms of the attitudinal variables, trust in AI companies was associated with significantly increased odds of choosing an AI provider (OR: 1.29, CI: 1.23–1.36, $p < .001$).

## Associations with race/ethnicity

As shown in Table 5 Model 3 and Fig 2, we found significant associations between AI uptake and respondent's race or ethnicity; specifically Black respondents had lower odds of selecting

**Table 6. Logistic Regression in Respondent Race/Ethnicity Subsets Predicting AI Provider Choice Versus Human Physician.**

|  | White |  | Black |  | Hispanic |  | Asian |  | Native |  |
|---|---|---|---|---|---|---|---|---|---|---|
| Illness Severity | 0.97 | (0.20) | 1.05 | (0.23) | 1.01 | (0.22) | 0.78 | (0.17) | 0.97 | (0.27) |
| Accuracy | 1.76** | (0.33) | 1.42+ | (0.28) | 1.68* | (0.36) | 1.69* | (0.37) | 0.82 | (0.22) |
| Incorporation | 0.97 | (0.18) | 0.86 | (0.17) | 0.81 | (0.17) | 0.85 | (0.18) | 0.70 | (0.18) |
| Established | 1.16 | (0.22) | 0.91 | (0.18) | 1.57* | (0.33) | 1.63* | (0.35) | 1.15 | (0.30) |
| Tailored | 1.11 | (0.21) | 1.33 | (0.27) | 1.09 | (0.23) | 1.14 | (0.25) | 0.92 | (0.24) |
| Listens | 1.25 | (0.24) | 1.45+ | (0.29) | 1.22 | (0.25) | 1.10 | (0.23) | 1.48 | (0.40) |
| Race Unbiased | 1.10 | (0.21) | 1.17 | (0.23) | 1.49+ | (0.32) | 0.85 | (0.18) | 0.95 | (0.25) |
| Financial Unbiased | 0.90 | (0.17) | 1.11 | (0.23) | 1.15 | (0.24) | 1.13 | (0.24) | 1.86* | (0.50) |
| Income | 1.00 | (0.03) | 1.02 | (0.04) | 1.07+ | (0.04) | 1.03 | (0.03) | 0.96 | (0.05) |
| Age | 1.00 | (0.01) | 0.99 | (0.01) | 0.99 | (0.01) | 0.99 | (0.01) | 0.99 | (0.01) |
| Conservative | 0.58* | (0.14) | 0.82 | (0.25) | 0.69 | (0.18) | 0.55* | (0.16) | 0.60+ | (0.18) |
| Religion Important | 0.59** | (0.12) | 0.43*** | (0.10) | 0.79 | (0.19) | 0.66+ | (0.15) | 0.80 | (0.24) |
| Sex | 0.81 | (0.16) | 1.43+ | (0.30) | 1.44+ | (0.32) | 0.68+ | (0.15) | 0.75 | (0.21) |
| Education | 1.10 | (0.08) | 1.13+ | (0.08) | 1.15+ | (0.09) | 1.03 | (0.09) | 1.31* | (0.15) |
| Married | 1.00 | (0.21) | 0.74 | (0.17) | 1.14 | (0.25) | 0.75 | (0.18) | 1.48 | (0.42) |
| Employed Fulltime | 1.02 | (0.22) | 1.13 | (0.26) | 1.07 | (0.25) | 1.41 | (0.34) | 0.62 | (0.21) |
| Good Health | 1.27 | (0.30) | 1.07 | (0.26) | 0.55* | (0.15) | 1.04 | (0.30) | 1.13 | (0.33) |
| Have Provider | 1.01 | (0.30) | 1.65+ | (0.46) | 0.76 | (0.21) | 1.01 | (0.30) | 1.54 | (0.64) |
| Trust Provider | 0.94 | (0.06) | 0.94 | (0.06) | 0.94 | (0.06) | 1.02 | (0.07) | 0.97 | (0.07) |
| Trust Hospital | 0.92 | (0.05) | 0.99 | (0.07) | 0.98 | (0.06) | 0.83* | (0.06) | 0.99 | (0.07) |
| Trust AI Companies | 1.36*** | (0.08) | 1.20** | (0.07) | 1.36*** | (0.08) | 1.36*** | (0.09) | 1.34*** | (0.09) |
| Order | 1.05 | (0.20) | 1.17 | (0.23) | 1.01 | (0.21) | 1.50+ | (0.32) | 0.99 | (0.27) |
| Illness Experience | 1.12 | (0.23) | 1.41 | (0.32) | 0.74 | (0.17) | 1.29 | (0.30) | 1.28 | (0.38) |
| N | 542 |  | 479 |  | 442 |  | 437 |  | 298 |  |

Notes: Odds Ratios; Standard errors in parentheses

+ *p < .10*

* *p < .05*

** *p < .01*

*** *p < .001. Income (16-point scale from less than $10,000 to more than $500,000), age (years), education (6-point scale from less than high school to graduate school), sex (1 = male, 0 = female). Likert-scales are coded as good health (1 = excellent/very good/good), religion viewed as important (1 = very/somewhat important), and conservative (very conservative/conservative).*

AI (OR = .73, CI: .55-.96, p = .023) and Native Americans had higher odds of selecting AI (OR: 1.37, CI: 1.01–1.87, p = .041) than White respondents.

Table 64 investigates the effects of other variables within race/ethnicity subsets. AI accuracy continued to be significant and positive regarding AI provider selection for Whites, Hispanics, and Asians. The PCP nudge towards AI as an established and easy option resulted in significantly greater odds of choosing AI for Hispanics and Asians, whereas learning that AI was unbiased by financial relationships with drugmakers and insurance companies increased AI selection for Native Americans. Interestingly, the racial subsets analyses revealed other demographic associations. Being conservative had the greatest effects on lowering AI selection for Whites and Asians, while the importance of religion significantly lowered odds for Whites and Blacks. For example, Black respondents who viewed religion as important have 57% less than equal odds of choosing the AI clinic, compared to those placing less importance on religion (OR: .43, CI: .27-.69, p = .001).

## Discussion

### Strengths and limitations

Our qualitative interview study with actual patients relies on a small, convenience sample in one city. As such it does not allow strong conclusions when standing alone. It does provide a groundwork for our rigorous quantitative approach, helping us to ensure that the vignettes will be clear and understandable, and also generating hypotheses subject to quantitative testing subsequently, through systematic manipulation of the vignettes.

Those hypotheses are tested in our randomized, blinded experiment, which allows causal inference about the impact of the manipulations, and the factorial design allows strong statistical power, because every respondent provides an observation for every variable. Our diverse population from a high-quality survey sample allows extrapolation to the U.S. population, when weighted.

Prior work has shown that vignette experiments can have external validity [33,34] but it may be harder for respondents to predict their own decisions in peculiar situations they have not experienced and where emotional salience is high. We speculate that in a real-world context, patients may be even more susceptible to their PCP's influence.

When it comes to race, we did find variations in preferences as to the AI system, but we do not suggest that these differences are likely to be biological or genetic in particular. Specifically, Black respondents were less likely to choose the AI system. Here, longstanding effects from structural racism could manifest in continued distrust. In contrast, Native Americans were more likely to choose the AI system, presenting a puzzle. Our Native American sample is older and more conservative than the national average for the group, and also had the smallest sample size of the racial groups, so we cannot rule out that this could be due to unmeasured variables affecting the outcome. Future studies with a larger sample could better shed light on this effect.

We purposefully included hypothetical features in our descriptions of the AI system, when testing a range of manipulations, such as its lack of racial bias or its proven accuracy beyond that of human specialists. However, these traits may or may not be true about any particular AI system.

While we tested two distinct clinical vignettes (leukemia and sleep apnea) with very different levels of severity and found similar results, other clinical situations may be different. Neither of these scenarios were presented as being acute or emergent, where patients may have greater deference to their PCP such situations [35,36,37]. Other manipulations or other implementations thereof may also have different effects.

Finally, in addition to the exclusions noted above, we had pre-registered a plan to also exclude respondents who were familiar with the condition (leukemia or sleep apnea, depending on experimental assignment) because of their work in a healthcare setting, or because they, or a close family or friend, has had it. However, a larger than expected portion of the sample turned out to be familiar with the conditions (N = 998) and these respondents were more familiar with sleep apnea ($N$ = 665) than leukemia ($N$ = 333). Excluding those respondents would unbalance the experimental conditions, and dramatically reduce statistical power, especially for racial/ethnic subgroups. Instead, we control for illness familiarity in the main models and include models that exclude illness familiarity in S1 Table and S2 Table. Results for experimental manipulations were similar, with AI accuracy, PCP nudging as established care, and AI listening to patients all significant predictors of uptake.

### Findings and implications

Consistent with other studies, we found substantial resistance to the use of AI, what some have called "algorithm aversion" [5] or "robophobia" [4]. Our study contributes to this literature by showing the robustness of this resistance to AI in a diverse population of patients and across a range of other manipulated features of and applications of an AI system.

We find this aversion to AI diagnosis across two levels of disease severity, within specific racial and ethnic groups, and even where the AI was proven to be more accurate and the PCP nudged the patient towards AI, though these efforts were significantly helpful. Some of our hypothesized attempts to mitigate this resistance were ineffectual.

Most surprising, having the PCP emphasize that he or she would explain and incorporate the AI system's advice into the ultimate treatment decision did not increase uptake of AI. We expected greater resistance to AI where the ultimately treatment decision is simply outsourced to this machine. In contrast, we expected that patients would be more accepting of medical AI in the half of the vignettes where AI was presented as merely an input to their trusted human physician's ultimate decision.

In a world where AI is (or will have the potential to be) actually more accurate than human specialists, our findings suggest that patients may suffer additional mortality and morbidity, and the healthcare system may suffer inefficiencies, due to patient resistance. The passage of time and expanded use of AI in a range of settings familiar to laypersons (such as self-driving cars) may help patients become more familiar with and supportive of AI in healthcare. Indeed, we found that generalized trust in AI companies is nearly as important for AI uptake as the disclosure that this particular AI is proven more accurate that human specialists. On the other hand, some sort of future scandal or crisis or politicization associated with AI may actually harm overall trust in such systems.

For simplicity, respondents were asked to assume insurance coverage and out-of-pocket costs would be the same for either clinic. Insurers or healthcare systems may someday use these economic dimensions to direct or undermine patient choice of provider, as a form of value-based reimbursement. Additionally, physicians or healthcare systems may deploy AI without giving the patient a choice or asking permission, especially where AI has become the standard of care. While such economic, professional, and managerial gatekeeping are common in a range of settings, they raise distinct ethical and legal concerns [38,39], recognizing that patient autonomy and patient welfare may sometimes come into tension. In addition to our focus on patient perspectives, additional research is required on drivers of physician uptake of AI.

Our findings will be useful for the development of theory-based and evidence-driven recommendations for how physicians and patients might integrate AI into the informed consent process and into real world use and delivery of care. Patient resistance to AI diagnosis may impinge uptake in ways that undermine treatment goals, but human physicians can support adoption, where the technology is designed with the patient experience in mind and supported by evidence of accuracy.

## Supporting information

**S1 Text. Yougov's Matching and Weighting Approach for The Sample Alternate Versions of Dependent Variable.**
(DOCX)

**S1 Table. Logistic Regression Predicting Choice of AI, Full Sample without Exclusions.**
(DOCX)

**S2 Table. Logistic Regression Predicting Choice of AI with All Exclusions (Pre-Registered Analysis).**
(DOCX)

## Author Contributions

**Conceptualization:** Christopher Robertson, Andrew Woods, Marvin J. Slepian.

**Data curation:** Andrew Woods, Kelly Bergstrand, Cayley Balser.

**Formal analysis:** Christopher Robertson, Kelly Bergstrand, Cayley Balser.

**Funding acquisition:** Christopher Robertson.

**Investigation:** Christopher Robertson, Andrew Woods, Kelly Bergstrand, Jess Findley.

**Methodology:** Christopher Robertson, Kelly Bergstrand, Marvin J. Slepian.

**Project administration:** Jess Findley, Cayley Balser.

**Supervision:** Christopher Robertson, Marvin J. Slepian.

**Writing – original draft:** Christopher Robertson, Andrew Woods, Jess Findley, Marvin J. Slepian.

**Writing – review & editing:** Christopher Robertson, Kelly Bergstrand, Jess Findley, Marvin J. Slepian.

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
