## [Decision Letter · Decision Letter 0]

22 Jun 2022

PDIG-D-22-00059

Diverse Patients’ Attitudes Towards Artificial Intelligence (AI) in Diagnosis

PLOS Digital Health

Dear Dr. Slepian,

Thank you for submitting your manuscript to PLOS Digital Health. Based on reviewer comments, we feel that it has merit but does not fully meet PLOS Digital Health's publication criteria as it currently stands. Therefore, we invite you to submit a revised version of the manuscript that addresses the points raised during the review process.

Please submit your revised manuscript within 30 days . If you will need more time than this to complete your revisions, please reply to this message or contact the journal office at digitalhealth@plos.org. Please include the following items when submitting your revised manuscript:

* A response letter that responds to various points raised by the editor and reviewer(s). You should upload this letter as a separate file labeled 'Response to Reviewers'.

We look forward to receiving your revised manuscript.

Kind regards,

Rutwik Shah, MD

Guest Editor

PLOS Digital Health

Journal Requirements:

a. Please clarify all sources of funding (financial or material support) for your study. List the grants (with grant number) or organizations (with url) that supported your study, including funding received from your institution. 

b. State the initials, alongside each funding source, of each author to receive each grant.

c. State what role the funders took in the study. If the funders had no role in your study, please state: “The funders had no role in study design, data collection and analysis, decision to publish, or preparation of the manuscript.”

d. If any authors received a salary from any of your funders, please state which authors and which funders.

2. Please send a completed 'Competing Interests' statement, including any COIs declared by your co-authors. If you have no competing interests to declare, please state "The authors have declared that no competing interests exist". Otherwise please declare all competing interests beginning with the statement "I have read the journal's policy and the authors of this manuscript have the following competing interests:"

You can also see our guidelines for your reference: https://journals.plos.org/digitalhealth/s/submission-guidelines#loc-competing-interests

3. In the online submission form, you indicated that your data will be submitted to a repository upon acceptance. We strongly recommend all authors deposit their data before acceptance, as the process can be lengthy and hold up publication timelines. Please note that, though access restrictions are acceptable now, your entire data will need to be made freely accessible if your manuscript is accepted for publication. This policy applies to all data except where public deposition would breach compliance with the protocol approved by your research ethics board. If you are unable to adhere to our open data policy, please kindly revise your statement to explain your reasoning and we will seek the editor's input on an exemption. Please be assured that, once you have provided your new statement, the assessment of your exemption will not hold up the peer review process.

4. Please provide separate figure files in .tif or .eps format and remove any figures embedded in your manuscript file. Please also ensure that all files are under our size limit of 10MB.

For more information about how to convert your figure files please see our guidelines: https://journals.plos.org/digitalhealth/s/figures

5. We notice that your supplementary materials are included in the manuscript file. Please remove them and upload them with the file type 'Supporting Information'. Please ensure that each Supporting Information file has a legend listed in the manuscript after the references list.

Additional Editor Comments (if provided):

Reviewers' comments:

Reviewer's Responses to Questions

**Comments to the Author**

1. Does this manuscript meet PLOS Digital Health’s publication criteria? Is the manuscript technically sound, and do the data support the conclusions? The manuscript must describe methodologically and ethically rigorous research with conclusions that are appropriately drawn based on the data presented.

Reviewer #1: Yes

Reviewer #2: Yes

Reviewer #3: Yes

2. Has the statistical analysis been performed appropriately and rigorously?

Reviewer #1: Yes

Reviewer #2: I don't know

Reviewer #3: Yes

3. Have the authors made all data underlying the findings in their manuscript fully available (please refer to the Data Availability Statement at the start of the manuscript PDF file)?

Reviewer #1: No

Reviewer #2: No

Reviewer #3: Yes

4. Is the manuscript presented in an intelligible fashion and written in standard English?

Reviewer #1: Yes

Reviewer #2: Yes

Reviewer #3: Yes

5. Review Comments to the Author

Reviewer #1: This interesting paper provides the results of a survey of a diverse sample of individuals regarding their likelihood of responding to the use of artificial intelligence in medical care. Working with a survey firm, the authors identified a sample with up-weighted representation designed to ensure sufficient representation of under-represented groups. Participants were asked to indicate whether they would follow the recommendation of an artificial intelligence-facilitated system, under various conditions of physician encouragement, indication of personalization, description of unbiased results, and other conditions. Results suggested a variety of attitudes differing across ethnic groups, political beliefs, religious beliefs, and others. 

This paper uses an interesting approach to address an important problem. By focusing on specific questions regarding types of patient responses, the survey addresses the key question of how patients might respond to specific scenarios. As the mixed results across multiple scenarios and groups suggest a complex variety of responses that might require further work to disentangle, this paper opens the door for substantial future work. 

As interesting as this paper might be, multiple revisions are needed to make this paper more appropriate for publication.

1. The paper mentions a qualitative study that was used to inform the subsequent survey, but few details are provided regarding the conduct and analysis of the study. Specific description of how the pre-study influenced the subsequent survey would be most useful. 

2. The use of weighting and unweighted analyses needs a bit of clarification. Given that the initial sample was weighted, it seems as if an "unweighted" analysis would involve correction of the collected sample back to represent underlying population distributions. It would help to define this analysis explicitly. 

3. The experimental conditions could use some clarifications. For accuracy, listens, tailored, racial, and financial, I assume that the difference is that the prompt is given or not. This should be clarified. 

4. The links between the text and the tabular results are a bit hard to interpret. It seems clear that the results in the first paragraphs under "effects of experimental manipulations" are tied to Table 3, but this isn't as clear as it might be. Clearer mention of specific results, etc. might be needed - for example, the description of "clearly listening" might be interpreted as relating to "tailored" or "listens". This should be clarified. Textual errors ("Error! reference source not found") exacerbate this difficulty. 

5. The data sharing statement does not adhere to current norms. Given that this is a survey, there's no reason why the exact scenarios, the survey response data, and the analysis code can't be published and shared publicly.

Reviewer #2: Thanks for the opportunity to read this article, where the authors present very interesting evidence on the acceptability and trustworthiness of AI in the clinical context. Here's some suggestions on parts of the paper that could be improved by the authors.

The introduction of the paper is nice and clear and does a good job to introduce and contextualise the current debate on AI in healthcare, but I thought that the authors could do a bit more to present their specific contribution on this issue. This is done partially at the start of the section on findings and implications, but could be anticipated at the end of the introduction. I think this could highlight more explicitly the key contribution of the author and how they add on current discussions.

The author explain how they've carried out both qualitative and quantitative research and they discuss the former as pre-study. I think it would be helpful if the authors could say more on why they went with both qualitative and quantitative and how the two types of results were integrated and weighted. This is partially discussed in the methodological section on statistical analysis, but I think it would be helpful if the authors could expand on the more general rationale behind their methodological choices. 

A more specific point on the findings of the authors, where I would suggest that the authors spend a bit more time, is about the specific role that the AI plays in the clinical vignettes. The authors mention towards the end of the paper that the inclusion of physician in treatment decision didn't effect uptake, but I wonder: are all situations/vignettes discussed by the authors such that the AI determines on its own the treatment decision? Is this a situation that can be easily compared to one where a physician relies on the AI and other tools? What do the authors think about differences across the spectrum? I wonder how much of the lack of acceptability of AI rests on deferring judgment and trust to machines completely, I wonder if similar issues of acceptability would show if judgment were fully deferred to PET scanning or other technologies. 

A final point where I would be interested to hear more from the authors concerns the concepts of acceptance and trustworthiness. What is the difference between the two, if any? Do the authors approach and use the two concepts as synonyms or are they the same? To me, it seems like acceptance is an issue that is less grounded on social interactions, which is rather the context where trust emerges. I mention this as a possible distinction because I wonder if the ethnic and socio-economic background of participants can have an effect on trustworthiness, rather than acceptance. I also wonder whether trustworthiness is more about the relationship established between users, infrastructures, technologies, practitioners, rather than the certainty and accuracy of the technology, which maybe plays a more important role for acceptance. 

See also recent work in philosophy of science on trust and trustworthiness as potential sources of material on these issues:

- https://doi.org/10.1093/bjps/axs007

- https://doi.org/10.1007/s11229-018-01913-z

Reviewer #3: In this paper, the authors conducted two experiments, interviews with patients in clinical vignettes and a survey from a diverse population, towards the selection of AI versus human physicians. The authors found that a primary care physician’s explanation about AI’s superior accuracy, a primary care physician’s gentle push towards AI, and a listening patient experience would increase the acceptance of AI in clinics. On the other hand, disease severity and other manipulations would not affect the acceptance of AI. Further, the preference of selecting AI in clinic varies among racial groups, Native Americans > European Americans > Africa Americans. Also, those population groups including senior group (with an older age), politically conservative, and viewing religion, are less likely to choose AI. 

The authors conducted the qualitive study with 24 patients (most of them are over 50 years old) from Tucson Arizona, and the quantitative survey using the internet. 

Qualitive study: 

Major: 

The sample size (a total of 24 patients, 10 Whites, 8 Hispanics, 3 Blacks, 2 Native Americans, and 1 Asian) collected in the study is relatively small to make solid conclusions.

Another weakness is the age distribution of these 24 patients, as mentioned in the paper, “ages ranged from 19 to 92 years, with most over 50”. The attitude of these patients may partially relate to the fact that AI was not significantly powerful/trustworthy back to 1970s. 

Make a concession, even the sample size is adequate, as mentioned in the paper “all patients are from Tucson, AZ”, it would be better to discuss and mention that the result and conclusion would be valid and appliable in the town of Tucson, Arizona.

Minor:

The definition of senior group (who were older), politically conservative, and viewing religion could be clearer. 

In the second paragraph of Page 6, it would be better to update some phrases. For examples, update “several” to “XX%” or xx out of 24, 

Quantitative study, 

Major: 

The race and ethnicity study reported a significant difference between race or ethnicity, black respondents had lower odds of selecting AI (OR=.73, CI:.55-.96, p=.023) and Native Americans had higher odds of selecting AI (OR: 1.37, CI: 1.01-1.87, p=.041) than White respondents. A major factor needs to be considered is the composition of each racial population group. As mentioned in the paper, each unit increase in education improves the chance of selecting AI providers. Therefore, compared with the white group, the lower odds of selecting AI in the black group may be due to the lower education level. Therefore, an in-depth study of the difference would be necessary to answer where it come from. Another factor is genetical difference between racial groups [1,2], which has been shown to have impact in different phenotypes. It would be better to discuss such point in the discussion section. 

Similarly, for other factors like sex, Religion Important, Age, etc., their group composition should be further investigated.

[1]: Martin, A.R., Kanai, M., Kamatani, Y., Okada, Y., Neale, B.M. and Daly, M.J., 2019. Clinical use of current polygenic risk scores may exacerbate health disparities. Nature genetics, 51(4), pp.584-591.

[2]: Gao, Y. and Cui, Y., 2020. Deep transfer learning for reducing health care disparities arising from biomedical data inequality. Nature communications, 11(1), pp.1-8.

6. PLOS authors have the option to publish the peer review history of their article (what does this mean?). If published, this will include your full peer review and any attached files.

**Do you want your identity to be public for this peer review?** For information about this choice, including consent withdrawal, please see our Privacy Policy.

Reviewer #1: Yes: Harry Hochheiser

Reviewer #2: No

Reviewer #3: No

---

## [Decision Letter · Decision Letter 1]

8 Feb 2023

PDIG-D-22-00059R1

Diverse Patients’ Attitudes Towards Artificial Intelligence (AI) in Diagnosis

PLOS Digital Health

Dear Dr. Slepian,

Thank you for submitting your manuscript to PLOS Digital Health. Based on the Reviewer 1 comments, there are still some questions to be answered and changes that are suggested. We feel that it has merit but does not fully meet PLOS Digital Health's publication criteria as it currently stands. Therefore, we invite you to submit a revised version of the manuscript that addresses the points raised during the review process.

Please submit your revised manuscript within 30 days (by 25th Feb 2023). If you will need more time than this to complete your revisions, please reply to this message or contact the journal office at digitalhealth@plos.org. Please include the following items when submitting your revised manuscript:

We look forward to receiving your revised manuscript.

Kind regards,

Rutwik Shah, MD

Guest Editor

PLOS Digital Health

Journal Requirements:

2. We have noticed that you have uploaded Supporting Information files, but you have not included a list of legends. Please add a full list of legends for your Supporting Information files after the references list. 

Additional Editor Comments (if provided):

Reviewers' comments:

Reviewer's Responses to Questions

**Comments to the Author**

1. If the authors have adequately addressed your comments raised in a previous round of review and you feel that this manuscript is now acceptable for publication, you may indicate that here to bypass the “Comments to the Author” section, enter your conflict of interest statement in the “Confidential to Editor” section, and submit your "Accept" recommendation.

Reviewer #1: (No Response)

Reviewer #2: All comments have been addressed

Reviewer #3: All comments have been addressed

2. Does this manuscript meet PLOS Digital Health’s publication criteria? Is the manuscript technically sound, and do the data support the conclusions? The manuscript must describe methodologically and ethically rigorous research with conclusions that are appropriately drawn based on the data presented.

Reviewer #1: Yes

Reviewer #2: Yes

Reviewer #3: Yes

3. Has the statistical analysis been performed appropriately and rigorously?

Reviewer #1: Yes

Reviewer #2: I don't know

Reviewer #3: Yes

4. Have the authors made all data underlying the findings in their manuscript fully available (please refer to the Data Availability Statement at the start of the manuscript PDF file)?

Reviewer #1: Yes

Reviewer #2: Yes

Reviewer #3: No

5. Is the manuscript presented in an intelligible fashion and written in standard English?

Reviewer #1: Yes

Reviewer #2: Yes

Reviewer #3: Yes

6. Review Comments to the Author

Reviewer #1: Thanks for the revisions to the paper. This revision is much

improved. However, there are a few lingering issue that should be

addressed.

The additional clarity regarding the qualitative analysis is

particularly useful. However, there is almost nothing said about how

the qualitative study influenced the subsequent survey. We are simply

told that "These materials were based on the vignettes that we tested

in the qualitative phase, with refinements and clarifications based on

feedback from those participants." More detail regarding these changes

is needed.

This shortcoming is directly related to a point raised by one of the

other reviewers: the lack of detail in the discussion of the

qualitative results. The authors are not inappropriate in mentioning

that the presentation of statistics might lead to some bias. However,

the use of terms such as "most" or "several" is arguably

worse. In this case, I believe the risk associated with quantifying

the results is acceptable, particularly as the bulk of the paper is

focused on the results of the qualitative study.

A few additional points came to mind in the re-reading of this paper:

1. Table 1a: The text presented to respondents is arguably quite

biased. The description of the AI case (Med-X) specifically mentions

blood and genetic analysis, whereas neither is mentioned in the

discussion of the specialist consult. Respondents might legitimately

read this as saying that the physician consult might not involve any

blood or genetic testing, and thus conclude that the physician's

decision might not be well-informed. 

2. There is a troubling statement in the introduction: "Although

initial forays into algorithmic decisions gave rise to similar

concerns (26), AI systems may be rigorously designed, tested, and

continuously monitored to minimize racial or financial biases in

healthcare (27, 28)." Although technically true, this statement lacks

important context - many, if not most, systems, are not built this

rigorously. As perceptions of rigor and considerations of bias might

be particularly important for responses to the survey, this limitation

should be acknowledged.

3. Statistical comparisons between groups should be added to figure

2B.

Reviewer #2: The authors have done a great job at addressing my concerns and suggestions in the first round of revisions, in particular concerning the initial presentation of their contribution, the rationale behind using quantitative and qualitative methods, differences between ethic backgrounds and acceptance rates, and the distinction between the different role of AI models in the vignettes.

Reviewer #3: Dear Authors:

Thank you for addressing my comments, and I have no more comments.

Best,

7. PLOS authors have the option to publish the peer review history of their article (what does this mean?). If published, this will include your full peer review and any attached files.

**Do you want your identity to be public for this peer review?** For information about this choice, including consent withdrawal, please see our Privacy Policy. 

Reviewer #1: Yes: Harry Hochheiser

Reviewer #2: No

Reviewer #3: No

---

## [Editor Report · Decision Letter 2]

20 Mar 2023

Diverse Patients’ Attitudes Towards Artificial Intelligence (AI) in Diagnosis

PDIG-D-22-00059R2

Dear Dr. Slepian,

We are pleased to inform you that your manuscript 'Diverse Patients’ Attitudes Towards Artificial Intelligence (AI) in Diagnosis' has been provisionally accepted for publication in PLOS Digital Health. After the second revision, the manuscript now satisfactorily addresses reviewer questions and comments. 

Best regards,

Rutwik Shah, MD

Guest Editor

PLOS Digital Health
